# Impact of Cardiac Injury on the Clinical Outcome of Children with Convulsive Status Epilepticus

**DOI:** 10.3390/children9020122

**Published:** 2022-01-18

**Authors:** Ahmed Ibrahim, Ahmed Megahed, Ahmed Salem, Osama Zekry

**Affiliations:** 1Department of Pediatrics, Faculty of Medicine, Suez Canal University, Ismailia 41511, Egypt; ahmansour_87@yahoo.com (A.M.); osamazekry@hotmail.com (O.Z.); 2Department of Cardiology, Faculty of Medicine, Suez Canal University, Ismailia 41511, Egypt; alsalem912@hotmail.com

**Keywords:** cardiac injury, convulsive status epilepticus, mortality, outcome

## Abstract

Objectives: the aim of this study was to determine the impact of cardiac injury on clinical profile, cardiac evaluation and outcome in patients hospitalized with convulsive status epilepticus (CSE). Materials and methods: this prospective observational study included 74 children with CSE. Cardiac injury was evaluated and defined using combination of cardiac troponin, electrocardiography (ECG) and echocardiography. Clinical outcome and mortality rates were compared in patients with and without cardiac injury. Results: A total of 74 patients with CSE were included in the study. Thirty-six (48.6%) patients demonstrated markers of cardiac injury. ECG changes occurred in 45.9% and echocardiographic signs of left ventricular systolic and diastolic dysfunction reported in 5.4% and 8.1%, respectively. The mean length of hospital stays and need for ICU admission were significantly higher in patients with cardiac injury compared to others. One third of patients with cardiac injury needed mechanical ventilation and this was significantly higher than patients without (*p* = 0.042). hypotension and/or shock developed in 25% of cardiac injury patients and most of them required inotropic support; this was significantly higher than others without markers of cardiac injury. The overall mortality in cardiac injury group was higher (13.9% vs. 2.6%); however, this difference was not statistically significant. Conclusion: Markers of cardiac injury were common and associated with poor clinical outcome and higher risk of mortality in patients with CSE, so extensive routine cardiovascular evaluation is essential in these patients.

## 1. Introduction

Convulsive status epilepticus (CSE) is the most common life-threatening pediatric neurological emergency, with high morbidity and mortality rates [1]. Previously, status epilepticus was defined as a seizure lasting more than 30 min [2]. However, more recently, status epilepticus is considered if a patient has continuous seizure activity for 5 min or recurrent seizures without regaining consciousness level in between [3].

Cardiac injury in CSE is associated with a combination of excessive catecholamine release, sympathetic overflow, and subsequent neurogenic myocardial stunning, resulting in subtle structural and functional myocardial damage with a high incidence of arrhythmia, stress-related cardiomyopathy, and heart failure [4]. Moreover, cardiac injury may be iatrogenic due to intravenous fluid overload and cardio-depressive effects of anti-seizure medications (ASMs) [5]. Most of the SE-induced deaths occurring within 30 days following seizure activity are attributed to lethal cardiac arrhythmia, pulmonary edema, hypotension, and circulatory collapse [6].

Cardiovascular-specific biomarkers have been identified as the most accurate indicators of myocardial infarction. Cardiovascular troponin-I (cTnI), in particular, is extremely selective for myocardial muscular tissue injury and is never produced following skeletal muscle injury [7]. The possibility of different release mechanisms for cTnI beyond the traditional myocardial ischemia causing acute coronary syndrome (ACS) was the conclusion of researchers who observed increased cTnI unrelated to ACS that occurred in multiple situations, such as sepsis, critical illness, and endocrine or neurologic disorders [8]. Electrocardiographic (ECG) changes are frequent during and after CSE, suggesting that these patients had high risk factors for cardiac stress and usually predicted high morbidity and mortality. These findings potentially change the method of monitoring and managing patients with SE [9]. Up to 50% of patients with CSE may develop myocardial stunting and stress-induced cardiomyopathy with characteristic reversible apical ballooning, ventricular dysfunction and less commonly wall motion abnormalities on echocardiographic assessment; all these structural and functional changes may worsen outcome and increase risk of mortality [10].

Following prolonged seizures, a considerable percentage of patients showed an increase in biomarkers of cardiac necrosis and dysfunction, which may indicate the onset of myocardial injury [11], but the prognostic value of this finding is not well defined.

Therefore, to address this gap in knowledge, this study aimed to investigate the prognostic value of myocardial damage biomarkers in patients hospitalized with CSE in terms of clinical profile, cardiac evaluation, and outcomes.

## 2. Materials and Methods

### 2.1. Study Design and Patient Population

This prospective observational study was conducted at the emergency department and pediatric intensive care unit of Suez Canal University Hospitals, Ismailia Governorate. It included 74 children admitted to the hospital for management of convulsive status epilepticus from March 2018 to June 2020.

All children aged 1 month to 12 years who presented with CSE were included in the study. CSE was defined as a child having generalized seizure lasting more than 5 min (10 min for focal seizure) or recurrent seizures without returning to the baseline in between [3]. The total duration of seizures was obtained from parental history as interviewed at the time of admission as well as from records of the referring doctors before and after admission. Refractory status epilepticus (RSE) was defined as continued CSE despite treatment with two ASMs [5].

Patients with any chronic illness other than epilepsy (hepatic, renal, endocrinal, or muscle disease); children with sepsis, congenital heart disease, myocarditis, myocardial disease, or arrhythmias; and children who survived cardiac surgery were excluded from the study.

Detailed histories were examined, including demographic data, seizure duration, etiology, seizure semiology according to the 2017 International League Against Epilepsy operational classifications [12], results of any electroencephalography (EEG) and neuroimaging, and type and number of anesthetic medications used. Complete physical examination, including blood pressure measurement, heart rate monitoring, temperature, detailed neurological examination, and ophthalmological assessment, was performed.

Laboratory investigations were performed in the first 6 h after admission to hospital. Initial routine laboratory investigations included complete blood count, liver function test, renal function test, blood sugar, serum calcium level, and arterial blood gases.

cTnI was measured using enzyme-linked immunoassay. Myocardial injury is diagnosed when cardiac troponin (cTn) levels exceed the 99th percentile upper reference limit (URL) according to the Fourth Universal Definition of Myocardial Infarction, 2018 [13].

### 2.2. Cardio-Functional Assessment: A Single Cardiologist Blinded to the Clinical and Laboratory Data of the Patients

Electrocardiography (ECG) was performed within 6 h of admission and repeated after 24 h. Herein, 12-lead ECGs were recorded at a standard 25 mm/s paper speed and gain setting of 10 mm/mV. ECGs were defined by identifying values for heart rate, QRS axis, PR interval, QTc interval (calculated using Bazett’s formula, prolonged QTc was defined as duration > 460 ms) [14], ST segment changes, T wave abnormalities, and voltage. Additionally, several forms of arrhythmias have also been described.

Echocardiography was performed within 24 h of admission. All the patients underwent standard transthoracic echocardiography using a General Electric Vivid 7 ultrasound system (GE Healthcare).

Standard echocardiography:

M-mode two-dimensional echocardiography was performed according to the recommendations of the American Society of Echocardiography [15]; left ventricular end systolic and diastolic dimensions were measured, the ejection fraction was obtained using Simpson’s rule, and the percentage of fractional shortening was calculated.

Tissue Doppler echocardiography:

Diastolic function was assessed in four-chamber views by measuring mitral flow velocities, early and late diastolic velocities (Em/Am ratio), and isovolumetric relaxation time (IVRT). The E/Em ratio was calculated as a preload-independent measure of filling pressure in the heart. Isovolumic contraction and relaxation durations are summed and divided by the ejection time to obtain the myocardial performance index (MPI), which is used to determine the global ventricular function [16].

### 2.3. Cardiac Injury Definition

Cardiac injury was defined as one or more of the following: (1) cardiac troponin ≥ 99th percentile URL, corrected for age and sex and (2) new-onset ECG changes categorized into ischemic changes, conduction abnormalities, or arrhythmias [17]. (3) Left ventricular systolic or diastolic dysfunction, which is defined as follows: systolic dysfunction if EF < 56% and/or FS < 28% [18], diastolic dysfunction is defined and classified according to the recommendations of the American Society of Echocardiography and the European Association of Cardiovascular Imaging [19].

### 2.4. Clinical Outcome Measures

(1)In hospital stay length/day(2)Inpatient transfer to intensive care unit (ICU)(3)Length of ICU admissions/day(4)Need for mechanical ventilation(5)Acute respiratory distress syndrome (ARDS), defined according to the Berlin definition [20](6)Hypotension/shock(7)Use of vasopressors(8)Acute kidney injury (AKI) is defined according to the Kidney Disease: Improving Global Outcomes (KDIGO) clinical practice guideline [21](9)In-hospital mortality

### 2.5. Statistical Analysis

Statistical analysis was performed using the Statistical Package for the Social Sciences (SPSS) version 19 software (SPSS Inc., Chicago, IL, USA). Data are presented as mean and standard deviation. Student’s *t*-tests were used to compare numerical data, while chi-square (χ2) or Fisher’s exact tests were used to compare categorical data. For all tests, *p* values < 0.05, were considered statistically significant.

## 3. Results

A total of 74 patients with CSE presented during the study period (43 males and 31 females), aged from 1 month to 12 years (mean age, 6.92 ± 2.34). The etiology of CSE was epilepsy-related in 27 (36.5%) patients, while acute symptomatic causes occurred in 18 (24.3%) patients, 11 (14.9%) had prolonged febrile seizures, and 10 (13.5%) cases were not identified (cryptogenic). The remaining 8 (10.8%) patients had remote symptomatic causes. Regarding seizure type, CSE was mainly generalized in 45 (60.8%) patients, while focal seizures were observed in 29 (39.2%) patients. The mean seizure duration was 28.6 ± 5.8 min and ranged from 5 to 96 min. Seizures were terminated after the administration of benzodiazepine in 12 (16.2%) patients. However, second- and third-line drugs were needed for 62 (83.8%) patients. Use of anesthetic medications (propofol, thiopental, and midazolam infusions) was required in 20 (27%) patients, and all of them needed ICU admission. They constituted 62.5% (20/32) of the total ICU admissions. The remainder were admitted because of respiratory compromise, cardiogenic shock, or for observation and continuous monitoring.

Thirty-six patients (48.6%) demonstrated cardiac injury markers. Seizure duration was significantly longer in patients with cardiac injury than in patients without cardiac injury (32.8 ± 6.4 vs. 24.7 ± 5.6, *p* = 0.001), and refractory seizure occurred in 18 (50%) children with cardiac injury compared to 10 (26.3%) in patients without cardiac injury (*p* = 0.036). There was a significant difference in the frequency of anesthetic use between the groups with and without markers of cardiac injury 13 (36.1%) vs. 7 (18.4%) respectively (*p* = 0.03). Children with CSE exhibited cardiac injury, independent of the seizure type, etiology, or number of ASMs used (Table 1).

Cardiac troponin levels were exceeded 99th centile (URL) in 28 patients (37.8%) with mean level among all patients 0.168 ± 0.08 ng/mL, in addition the mean concentration of cardiac troponin was significantly different among patients with and without cardiac injury (0.24 ± 0.14 vs. 0.09 ± 0.03 ng/mL, *p* = 0.001) respectively (Table 1).

ECG changes occurred during the 1st 24 h of CSE in 34 patients (45.9%). Cardiac arrhythmias occurred in 15 patients (20.3%), the most frequent abnormality was premature ventricular contractions in 5 patients (6.7%), 3 patients (4%) had premature atrial contractions, and four patients (5.4%) had sinus bradycardia, while one patient developed life-threatening ventricular fibrillation requiring resuscitative measures; all types of arrhythmias resolved spontaneously except three patients required intervention. Conduction abnormalities occurred in 10 patients (13.5%), QTc was prolonged in 4 patients (5.4%), bundle branch block developed in 5 patients (6.7%), and atrioventricular block in one patient. Ischemic pattern was detected in 9 patients (12.2%), including T wave changes in 4 patients (5.4%), ST segment depression in 3 patients (4%), and ST segment elevation in one patient, while a new Q wave developed in one patient.

Echocardiography performed on all patients with CSE revealed overall left ventricular systolic dysfunction in 4 patients (5.4%), while left ventricular diastolic dysfunction was detected in 6 patients (8.1%). EF was significantly lower in patients with cardiac injury than in those without cardiac injury (62.32 ± 6.87 vs. 66.56 ± 5.43, *p* = 0.03), while other standard echocardiographic parameters by M-mode were not statistically significant between the two groups. Using standard Doppler and tissue Doppler imaging studies, we found that the mitral E/A ratio and MPI were significantly elevated in patients with cardiac injury (1.62 ± 0.21, 0.42 ± 0.10, respectively) compared to those without cardiac injury (1.24 ± 0.18, 0.34 ± 0.16, respectively) (Table 2).

The mean length of hospital stay in days was significantly longer among patients with cardiac injury than among those without cardiac injury (13.6 ± 3.6 vs. 11.9 ± 2.9, *p* = 0.028). Twenty patients with cardiac injury (55.6%) required transfer to the ICU compared to 12 (31.6%) in patients without cardiac injury, and the length of ICU stay was significantly higher in patients with cardiac injury than in others (5.6 ± 2.2 vs. 3.6 ± 1.8, *p* = 0.001).

One-third of patients with cardiac injury required mechanical ventilation compared to 13.2% of non-cardiac injury patients.

Patients who developed ARDS (8.3% vs. 2.6%) and AKI (19.4% vs. 15.8%) were not significantly different between the two groups. Hypotension and/or shock occurred in 9 (25%) patients with cardiac injury, and two-thirds of them required inotropic support, which was significantly higher than that in patients without cardiac injury.

Five deaths (13.9%) were identified within the follow-up period in the cardiac injury group compared to one patient (2.6%) without markers of cardiac injury; however, this difference was not statistically significant (*p* = 0.076). Mortality was due to anoxic brain damage in three cases, two patients due to sudden cardiac arrest, and one patient due to ARDS (Table 3).

## 4. Discussion

We found that cardiac injury markers occur in nearly half of patients with CSE, which may be attributed to the interaction of sympathetic overdrive, catecholamine toxicity, treatment complications, and myocardial ischemia resulting in cardiac fibrosis, arrhythmias, and stress (Takotsubo) cardiomyopathy [22]. In our patients, long SE duration and RSE were significantly correlated with cardiac injury. Hocker et al. [5] demonstrated that two-thirds of patients with RSE have markers of cardiac injury; however, because cardiac assessment was inconsistent and lacking, the sample was biased with severe RSE patients, these findings do not estimate the real prevalence of cardiac abnormalities in these patients. In patients who have suffered prolonged seizure duration, continuous epileptic discharges are thought to propagate to the central autonomic network, altering or disrupting the normal autonomic regulation of essential cardiac functions and causing subendocardial ischemia [23]. In addition, many anesthetic medications, particularly propofol and barbiturates, may be associated with cardiorespiratory depression, as well as peripheral vasodilatation, leading to profound hypotension, and their potential role cannot be neglected [5].

In our study, cardiac troponin levels were elevated in 28 patients (37.8%). Consistent with our findings, Nass et al. [24] reported that subclinical cardiac stress associated with generalized convulsive seizure (GCS) was found in approximately 25% of patients using more sensitive biomarkers, such as high-sensitivity troponin T (hs TnT). This may indicate that a critical amount of damage has accrued with repeated GCS and may reflect cardiac microinjuries with potentially dangerous consequences in the heart. Postictal cardiac troponinemia has been reported following generalized seizures with subtle myocardial tissue damage, especially in patients with cardiovascular risk factors [25]. Duchnowski and colleagues reported that the hs TnT is a reliable predictor of cardiogenic shock necessitating mechanical circulatory support. Moreover, elevated troponin levels were associated with an increased risk of sudden cardiac arrest [26,27]. In contrast to our findings, Mehrpour et al. [28] found no evidence of an increase in cTnI serum levels in patients with a healthy cardiovascular system following generalized CSE episodes. However, further evaluation should be performed if there are any postictal troponin elevations, since they might indicate myocardial damage caused by neurocardiogenic processes or primary cardiac causes [29].

In the present study, cardiac arrhythmias occurred in 15 patients (20.3%). El Amrousy et al. [30] and Kurukumbi et al. [9] reported similar results in both children and adults. Activated sympathetic tone causes damage to cardiac myofilaments and arrhythmogenic changes in cardiac electrical activity, as well as an increased risk of ventricular arrhythmias [31]. Ischemic ECG changes occurred in 9 patients (12.2%), mostly with ST segment changes and T wave inversion, and it is probable that cardiac oxygen deprivation results from CSE, which is caused by massive muscle activity, respiratory depression, and noticeable, long-lasting sinus tachycardia. Temporary ECG abnormalities, such as ST-segment depression and T-wave inversion, may indicate myocardial ischemia, in keeping with this hypothesis [32].

Echocardiographic findings in CSE demonstrated subtle left ventricular systolic and diastolic dysfunction. Catecholaminergic toxicity, microvascular ischemia, and anesthetic drugs are believed to be potential causes, leading to structural and functional myocardial alterations and a heart failure-like phenotype.

Patients with markers of cardiac injury had longer hospital stays, and more than half of them required ICU admission. Myocardial stunning from massive catecholamine release (Takotsubo cardiomyopathy) may affect up to half of these patients with a higher rate of complications, respiratory failure secondary to pulmonary edema, hypotension, cardiac arrhythmias, and less likely cardiogenic shock [33], which requires a longer hospital stay and ICU transfers.

Our study demonstrated that one-third of patients with cardiac injury needed mechanical ventilation compared to 13.2% of non-cardiac injury patients. More than 21% of those with SE require endotracheal intubation, which is more common among the elderly or those with refractory seizures, pulmonary edema, respiratory depression, repeated seizures, or low Glasgow Coma Scale are all reasons for intubation in status epilepticus patients. [34]. Acute catecholaminergic surge with subsequent myocardial damage increases left ventricular pressure, resulting in neurocardiogenic pulmonary edema. Furthermore, sympathetic overactivity stimulates adrenergic receptors and increases pulmonary vascular endothelial permeability [35].

We found that hypotension and/or shock occurred in 25% of cases with cardiac injury, and two-thirds of them required inotropic support, which was significantly higher than that in patients without markers of cardiac injury. Belcour et al. [36] reported that more than half of patients admitted to the ICU with CSE developed stress cardiomyopathy, which is clinically associated with hypotension and low cardiac output states, and more frequently required inotropic support compared to patients without cardiomyopathy. Seizure-associated Takotsubo syndrome is characterized by a rapid decline in blood pressure and hemodynamic status, either as a result of pump failure or arrhythmia [37].

The overall mortality rate was higher among patients with cardiac injury (13.9%) than among those without cardiac injury (2.6%), confirming the role of cardiovascular complications as a cause of death in patients with CSE. Lethal cardiac arrhythmias with or without gross structural cardiac damage are one possible mechanism leading to death following SE [38]. In an animal model of SE, myocardial ischemia and sympathetic over-stimulation cause modest damage to cardiac myofilaments, resulting in electrical alterations in heart function and an increased risk of fatal arrhythmias [31]; Kurukumbi et al. [9] reported that all deaths due to CSE had ECG abnormalities; therefore, strict monitoring of cardiac function is critical in the management of SE. Furthermore, seizure-related cardiac dysfunction is thought to be an important pathophysiological mechanism in sudden cardiac death [39].

The novelty of this study was the use of multimodal assessment tools to define cardiac injury in a pediatric population with CSE. A combination of these evaluation methods was useful in stratifying patients at risk in emergency settings.

Our study had some limitations. First, the cardiovascular evaluation of subjects prior to admission could not be obtained, so we could not rule out the possibility that the patients’ cardiac abnormalities had already existed before the CSE. Second, all investigations were performed only in the first 24 h after seizures, so subsequent abnormalities could not be detected. Third, most of our patients were treated with anesthetic drugs, so it is difficult to determine whether seizures or drugs were the cause of cardiac dysfunction. Fourth, the underlying causes of seizures might have affected the cardiac function of our patients.

## 5. Conclusions

Markers of cardiac injury are common, occurring early in patients with CSE, and appear to be an effective risk stratification tool for CSE patients, as well as useful in predicting the clinical outcome of these patients. We believe that these patients can be effectively evaluated using the noninvasive techniques described in this study during regular follow-up periods. If necessary, additional care and early intervention can be provided to reduce the risk of morbidity and mortality. Future studies with larger sample sizes are needed to utilize standardized cardiac assessment procedures to detect early cardiac changes and predict patient outcomes.

## Figures and Tables

**Table 1 children-09-00122-t001:** Clinical and laboratory characteristics of patients with and without cardiac injury.

Parameters	With Cardiac Injury (*n* = 36)	Without Cardiac Injury (*n* = 38)	*p*-Value
Age (years) Mean ± SD	7.2 ± 2.46	6.8 ± 2.24	0.466
Sex (No., %)			
Males	22 (61.1%)	21 (55.2%)	0.609
Females	14 (38.9%)	17 (44.8%)	
Seizure duration (minutes) Mean ± SD	32.8 ± 6.4	24.7 ± 5.6	0.001 *
Predominant seizure semiology (No., %)			
Focal	13 (36.1%)	16 (42.1%)	0.599
Generalized	23 (63.9%)	22 (57.9%)	
Refractory seizure (No., %)	18 (50%)	10 (26.3%)	0.036 *
Seizure etiology (No., %)			
Prolonged febrile seizure	5 (13.9%)	6 (15.8%)	0.819
Acute symptomatic cause (CNS infection, stoke, head injury, drug induced)	10 (27.8%)	8 (21%)	0.498
Epilepsy related	12 (33.3%)	15 (39.5%)	0.582
Remote symptomatic	3 (8.3%)	5 (13.2%)	0.50
Cryptogenic (absence of known acute or remote etiology)	6 (16.7%)	4 (10.5%)	0.438
Number of antiseizure medications used	2.8 ± 1.2	2.4 ± 1.4	0.192
cTnI (ng/mL)	0.24 ± 0.14	0.09 ± 0.03	0.001 *

* Indicates significant difference between groups.

**Table 2 children-09-00122-t002:** Echocardiographic parameters in patients with and without cardiac injury.

Echocardiographic Parameters	With Cardiac Injury (*n* = 36)	Without Cardiac Injury (*n* = 38)	*p*-Value
LVED dimension (mm)	36.86 ± 5.24	34.54 ± 5.68	0.072
LVES dimension (mm)	24.45 ± 4.28	23.24 ± 3.78	0.20
Fractional shortening (%)	33.72 ± 5.84	35.74 ± 4.86	0.10
Ejection fraction (%)	62.23 ± 6.87	66.56 ± 5.43	0.03 *
Septum thickness (mm)	7.32 ± 1.45	7.21 ± 1.34	0.735
Early mitral inflow velocity (E) (cm/s)	91.26 ± 4.56	86.89 ± 2.71	0.124
Late mitral inflow velocity (A) (cm/s)	62.24 ± 7.98	58.86 ± 3.34	0.046 *
E/A ratio	1.62 ± 0.21	1.24 ± 0.18	0.001 *
E/e′ ratio	8.78 ± 2.42	8.45 ± 2.23	0.543
IVRT (ms)	54.32 ± 8.28	51.62 ± 7.46	0.056
MPI	0.42 ± 0.10	0.34 ± 0.16	0.012 *

* Indicates significant difference between groups. LEVD, left ventricular end-diastolic; LVES, left ventricular end-systolic; E/e′, early mitral inflow velocity/early diastolic mitral annular velocity; IVRT, isovolumetric relaxation time; MPI, myocardial performance index.

**Table 3 children-09-00122-t003:** Outcome parameters in patients with and without cardiac injury.

Outcome Parameters	With Cardiac Injury (*n* = 36)	Without Cardiac Injury (*n* = 38)	*p*-Value
In hospital stay length/day	13.6 ± 3.6	11.9 ± 2.9	0.028 *
Inpatient transfer to intensive care unit (ICU)	20 (55.6%)	12 (31.6%)	0.038 *
Length of ICU admissions/day	5.6 ± 2.2	3.6 ± 1.8	0.001 *
Need for mechanical ventilation	12 (33.3%)	5 (13.2%)	0.042 *
Acute respiratory distress syndrome (ARDS)	3 (8.3%)	1 (2.6%)	0.284
Hypotension/shock	9(25%)	3 (7.9%)	0.047 *
Use of vasopressors	6 (16.7%)	1 (2.6%)	0.039 *
Acute kidney injury (AKI)	7 (19.4%)	6 (15.8%)	0.686
In-hospital mortality	5 (13.9%)	1 (2.6%)	0.076

* Indicates significant difference between groups.

## Data Availability

Datasets are available on request.

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
