# Peer review of "Impact of Cardiac Injury on the Clinical Outcome of Children with Convulsive Status Epilepticus"

_children, 2022, doi:10.3390/children9020122_

Round 1

Reviewer 1 Report

Thank you for the opportunity to review your manuscript entitled " Impact of cardiac injury on the clinical outcome of children with convulsive status epilepticus ".

Troponin T is a protein forming part of the contractile apparatus of the striated muscle. The function of TnT in all types of striated muscles is the same, but cTnT is different from TnT found in skeletal muscles. Therefore, cTnT detected in plasma is a highly specific marker of myocardial damage (necrosis). High -sensitivity troponin tests, available for the past several years, detect troponin levels with a high degree of credibility (1).

Cardiogenic shock and sudden cardiac arrest are a very serious complications. Mechanical circulatory support is a recognized method of treating patients with these complications. cTnT can be used to predict a cardiogenic shock requiring mechanical circulatory suport and sudden cardiac arrest (2,3).

Was cardiac Troponin T (cTnT) assessed also in the presented study?

It is worth quoting the articles below:

  1. doi: 10.20452/pamw.4107
  2. doi: 10.1097/SHK.0000000000001360
  3. doi: 10.5603/CJ.a2019.0005

The article presented for review is well written and brings new information.
The aim of the study is clear.

The research question is clearly outlined.

The process of subject selection is clear. The variables are defined and measured appropriately. The study methods are valid and reliable. There is enough detail in order to replicate the study.

The results are discussed from multiple angles and placed into context without being overinterpreted. The conclusions answer the aims of the study. The conclusions supported by references and results. The limitations of the study are opportunities to inform future research.
Overall.

Author Response

Response to Reviewer 1 Comments

Point 1: Was cardiac Troponin T (cTnT) assessed also in the presented study?

Response 1: In our study cardiac troponin I (cTnI) is only measured, together with other parameters of cardiac injury (ECG and echocardiography)

Point 2: It is worth quoting the articles below:

doi: 10.20452/pamw.4107

doi: 10.1097/SHK.0000000000001360

doi: 10.5603/CJ.a2019.0005

Troponin T is a protein forming part of the contractile apparatus of the striated muscle. The function of TnT in all types of striated muscles is the same, but cTnT is different from TnT found in skeletal muscles. Therefore, cTnT detected in plasma is a highly specific marker of myocardial damage (necrosis). High -sensitivity troponin tests, available for the past several years, detect troponin levels with a high degree of credibility (1).

Cardiogenic shock and sudden cardiac arrest are a very serious complications. Mechanical circulatory support is a recognized method of treating patients with these complications. cTnT can be used to predict a cardiogenic shock requiring mechanical circulatory suport and sudden cardiac arrest (2,3).

Response 2: Thank you very much for this kind comment. We believe that this comment helped us in improving our manuscript.

Two suggested references were added in the manuscript

Regarding this reference “doi: 10.20452/pamw.4107” we caould not able to add

“Troponin T is a protein forming part of the contractile apparatus of the striated muscle. The function of TnT in all types of striated muscles is the same, but cTnT is different from TnT found in skeletal muscles. Therefore, cTnT detected in plasma is a highly specific marker of myocardial damage (necrosis). High -sensitivity troponin tests, available for the past several years, detect troponin levels with a high degree of credibility (1).”

For these reasons:

  • In our study we measured mainly cTnI
  • The following paragraph was conducted in the introduction section and carry the same meaning. So we try to avoid duplication of information

“Cardiovascular-specific biomarkers have been identified as the most accurate indicators of myocardial infarction. Cardiovascular troponin-I (cTnI), in particular, is ex-tremely selective for myocardial muscular tissue injury and is never produced following skeletal muscle injury (7).”

This valuable paragraph is added to the discussion section to determine the predictive role of cardiac troponins of circulatory collapse and SCA

Duchnowski and colleagues reported that the high-sensitivity troponin T is a reliable predictor of cardiogenic shock necessitating mechanical circulatory support. Moreover, elevated troponin levels were associated with an increased risk of sudden cardiac arrest (1,2)

  • Duchnowski P, Hryniewiecki T, KuÅ›mierczyk M, et al. High-Sensitivity Troponin T Predicts Postoperative Cardiogenic Shock Requiring Mechanical Circulatory Support in Patients With Valve Disease. Shock. 2020;53(2):175-178.
  • Duchnowski P, Hryniewiecki T, KuÅ›mierczyk M, et al. Postoperative high-sensitivity troponin T as a predictor of sudden cardiac arrest in patients undergoing cardiac surgery. Cardiol J. 2019;26(6):777-781.

Finally, we are grateful for the valuable comments and suggestions provided

Reviewer 2 Report

Dear Authors,

the manuscript reports interesting data on the cardiac injuries of patients with convulsive status epilepticus.
But it is necessary that the data relating to the clinical characteristics of the status be reported in more detail, with a more detailed correlation between the parameters indicative of the severity of the status and the presence and severity of cardiac injuries.

I would like to point out the lack of:
• number of patients with initial status (response to first drug, I suppose benzodiazepine), defined, refractory;
• clarifications on the measurement of the duration of the status: from the start of the observation? from the start of treatment? based on the medical history?
• reasons for hospitalization in ICU and / or mechanical ventilation. How many patients thus treated for refractory status therapy (were all patients with refractory status admitted to the ICU?) and how many for other complications?
• data on the drugs used; in particular, how many patients have been treated with anesthetic drugs (propofol, midazolam, thiopentone...)? all patients with refractory status?

In the text (lines 147-149) the etiology of CSE is reported only for 66 patients (missing "remote symptomatic").

The citation of the Report of the ILAE Task Force on Classification of Status Epilepticus (reference 12) seems appropriate to me also previously (line 34) in association with reference 3.

Author Response

Response to Reviewer 2 Comments

the manuscript reports interesting data on the cardiac injuries of patients with convulsive status epilepticus.

We are grateful for the detailed and positive comments and suggestions provided by reviewer.

Point 1: But it is necessary that the data relating to the clinical characteristics of the status be reported in more detail, with a more detailed correlation between the parameters indicative of the severity of the status and the presence and severity of cardiac injuries.

Response 1: Thank you very much for your kind comment. We believe that this comment helped us in re-writing our manuscript to make it different and unique from other already published papers. We focused more. We nearly re-wrote the result part to avoid the problems mentioned by the reviewer. We focused more on the details of the clinical characteristics of status and correlation with cardiac injury. We added several paragraphs to the results and discussion section to explain this point.

These points will be discussed one by one

Point 2: number of patients with initial status (response to first drug, I suppose benzodiazepine), defined, refractory;

Response 2: Seizures were terminated after the administration of benzodiazepine in 12 (16.2%) patients. However, second- and third-line drugs were needed for 62 (83.8%) patients. Use of anesthetic medications (propofol, thiopental, and midazolam infusions) was required in 20 (27%) patients, and all of these patients needed ICU admission.

Point 3:  clarifications on the measurement of the duration of the status: from the start of the observation? from the start of treatment? based on the medical history?

Response 3:

This paragraph added to the Materials and methods section “The total duration of seizures was obtained from parental history as interviewed at the time of admission as well as from records of the referring doctors before and after admission.” To clarify method of measurement of seizure duration

This statement added to the result section “The mean seizure duration was 28.625.8 minutes and ranged from 5 to 96 minutes.”

Point 4: reasons for hospitalization in ICU and / or mechanical ventilation. How many patients thus treated for refractory status therapy (were all patients with refractory status admitted to the ICU?) and how many for other complications?

Response 4:

Seizures were terminated after the administration of benzodiazepine in 12 (16.2%) pa-tients. However, second- and third-line drugs were needed for 62 (83.8%) patients. Use of anesthetic medications (propofol, thiopental, and midazolam infusions) was required in 20 (27%) patients, and all of them needed ICU admission. They constituted 62.5% (20/32) of the total ICU admissions. The remainder were admitted because of respiratory compromise, cardiogenic shock, or for observation and continuous monitoring.

Point 5: data on the drugs used; in particular, how many patients have been treated with anesthetic drugs (propofol, midazolam, thiopentone...)? all patients with refractory status?

Response 5: Use of anesthetic medications (propofol, thiopental, and midazolam infusions) was required in 20 (27%) patients and all of these patients needed ICU admission.

Point 6: In the text (lines 147-149) the etiology of CSE is reported only for 66 patients (missing "remote symptomatic").

Response 6: we wrote the missed data

Point 7: The citation of the Report of the ILAE Task Force on Classification of Status Epilepticus (reference 12) seems appropriate to me also previously (line 34) in association with reference 3.

Response 7: we omitted reference no. 3 and we made the citation of the Report of the ILAE Task Force on Classification of Status Epilepticus (reference 12) for both paragraphs

Point 8: But it is necessary that the data relating to the clinical characteristics of the status be reported in more detail, with a more detailed correlation between the parameters indicative of the severity of the status and the presence and severity of cardiac injuries.

Response 8: We described the correlation between cardiac injury and seizure duration, RSE, use of anesthetic drugs, seizure type, number of ASMs and seizure etiology  

Thirty-six patients (48.6%) demonstrated cardiac injury markers. Seizure duration was significantly longer in patients with cardiac injury than in patients without cardiac injury (32.8±6.4 vs 24.7±5.6, p=0.001), and refractory seizure occurred in 18 (50%) children with cardiac injury compared to 10 (26.3%) in patients without cardiac injury (p=0.036). There was a significant difference in the frequency of anesthetic use between the groups with and without markers of cardiac injury 13 (36.1%) vs 7 (18.4%) respectively (p = 0.03). Children with CSE exhibited cardiac injury, independent of the seizure type, etiology, or number of ASMs used.

Also we discussed significant correlations in the discussion section

In our patients, long SE duration and RSE were significantly correlated with cardiac injury. Hocker et al. (5) demonstrated that two-thirds of patients with RSE have markers of cardiac injury; however, because cardiac assessment was inconsistent and lacking, the sample was biased with severe RSE patients, these findings do not estimate the real prevalence of cardiac abnormalities in these patients. In patients who have suffered prolonged seizure duration, continuous epileptic discharges are thought to propagate to the central autonomic network, altering or disrupting the normal autonomic regulation of essential cardiac functions and causing subendocardial ischemia (23). In addition, many anesthetic medications, particularly propofol and barbiturates, may be associated with cardiorespiratory depression as well as peripheral vasodilatation leading to pro-found hypotension, and their potential role cannot be neglected (5).

Round 2

Reviewer 2 Report

Dear Authors,

in my opinion, the quality of your paper has improved with the additions you made in the correction of the first version. My compliments.

A few points remain to be clarified:

- line 154: “seizure duration was 28.625.8 minutes” needs correction;

- refractory seizures: occurred in 28 patients (18+10) (lines 163-164 and table 1), but in the text (lines 156-157) only 20 patients needed anesthetic medications. Does this mean that there were patients with refractory status who were not treated with anesthetic medications? The contradiction needs correction or explanation.

- ST segment depression: in 4 patients (line 182), but only 3 in table 2. (In general the first part of table 2 - ECG abnormalities, 18 lines - repeats data presented in the text and obviously does not require a comparison with patients without cardiac injury. Therefore, evaluate the opportunity to report in table 2 only the comparison of the echocardiographic parameters in the two populations).

- Acute kidney injury in patients with cardiac injury: 19.4% in table 3, but 19.7% in the text (line 201).

Best regards.

Author Response

Response to Reviewer 2 Comments

in my opinion, the quality of your paper has improved with the additions you made in the correction of the first version. My compliments.

Thank you very much for your kind comments and suggestions . We believe that these comment helped us in improving our manuscript.

Point 1: line 154: “seizure duration was 28.625.8 minutes” needs correction;

Response 1: Thank you for exact observation, we corrected it in the revised manuscript (28.6±5.8)

Point 2: refractory seizures: occurred in 28 patients (18+10) (lines 163-164 and table 1), but in the text (lines 156-157) only 20 patients needed anesthetic medications. Does this mean that there were patients with refractory status who were not treated with anesthetic medications? The contradiction needs correction or explanation.

Response 2: Thank you very much for the very kind comment,

In the materials and methods section we define RSE as “Refractory status epilepticus (RSE) was defined as continued CSE despite treatment with two ASMs (5).

And according to this definition we had 28 patients fulfilled this criteria, only 20 of them needed anesthetic agents either propofol, thiopental, or midazolam infusions.

The remaining 8 patients, seizures not terminated with two ASMs and need third drug which is not belong to the anesthetic medications group either (Levitracetam or Valproate)

And to avoid over-interpretations of our results and to keep it concise we did not mention all these details.

Point 3: ST segment depression: in 4 patients (line 182), but only 3 in table 2.

Response 3: corrected in the revised manuscript

Point 4: (In general the first part of table 2 - ECG abnormalities, 18 lines - repeats data presented in the text and obviously does not require a comparison with patients without cardiac injury. Therefore, evaluate the opportunity to report in table 2 only the comparison of the echocardiographic parameters in the two populations).

Response 4: we are grateful for this suggestion. We agree with the reviewer about his idea. So we deleted the first section in table 2 (ECG abnormalities in patients with cardiac injury) and it was sufficient to mention the details in the text.

This valuable idea is needed to avoid repetition of data and to keep integrity and coherence of tables

Point 5: Acute kidney injury in patients with cardiac injury: 19.4% in table 3, but 19.7% in the text (line 201).

Response 5:  We are sorry about this mistake. We corrected it in the revision form.
